# Copy Number Variations of *KLF6* Modulate Gene Transcription and Growth Traits in Chinese Datong Yak (*Bos Grunniens*)

**DOI:** 10.3390/ani8090145

**Published:** 2018-08-21

**Authors:** Habtamu Abera Goshu, Xiaoyun Wu, Min Chu, Pengjia Bao, Xuezhi Ding, Ping Yan

**Affiliations:** 1Key Laboratory of Yak Breeding Engineering, Lanzhou Institute of Husbandry and Pharmaceutical Sciences, Chinese Academy of Agricultural Science, Lanzhou 730050, China; wuxiaoyun@caas.cn (X.W.); chumin@caas.cn (M.C.); baopengjia@caas.cn (P.B.); dingxuezhi@caas.cn (X.D.); 2Oromia Agricultural Research Institute, Bako Agricultural Research Center, P.O. Box 03, Bako, Ethiopia

**Keywords:** association, gene expression, transcription, quantitative traits, yak population

## Abstract

**Simple Summary:**

Associating copy number variations (CNVs) with growth and gene expression are a powerful approach for identifying genomic characteristics that contribute to phenotypic and genotypic variation. A previous study using next-generation sequencing illustrated that the *KLF6* gene resides within CNV regions (CNVRs) of yak populations that overlap with quantitative trait loci (QTLs) of meat quality and growth. As a result, this research aimed to assess the prevalence of *KLF6* CNVs in five Chinese domestic yak breeds and to determine their association with growth and gene expression level. These results confirmed the significant function of *KLF6* CNVRs in determining the mRNA expression levels of this gene in *Bos grunniens* skeletal muscles and the expression of this gene affects quantitative growth traits in yak populations through the negative relationship between DNA copy numbers and gene expression. This work provides the first observation of the biological role of *KLF6* CNVs in Chinese Datong yak breeds and might therefore provide a novel opportunity to utilize data on CNVs in designing molecular markers for the selection of animal breeding programs for larger populations of various yak breeds.

**Abstract:**

Copy number variation (CNV) is a significant marker of the genetic and phenotypic diversity among individuals that accounts for complex quantitative traits of phenotype and diseases via modulating gene dosage and disrupting coding regions in the genome. Biochemically, Kruppel-like factor 6 (*KLF6*) genes plays a significant role in the regulation of cell differentiation and proliferation and muscle development. The aim of this study was to detect the distributions of *KLF6* copy number variations (CNVs) in five breeds of domestic yak and to explore their effect on growth traits and gene expression. The data were analyzed by real-time quantitative PCR (qPCR). Our results elucidated that a decreased CNV in the *KLF6* gene is more highly associated (*p* < 0.05) with various growth traits than increased or normal CNVs in six-month-old and five-year-old Datong yak. Nevertheless, negative correlations between the DNA copy number and *KLF6* gene expression were observed in the skeletal muscle of adult Datong yak. These results suggest that CNVs of the *KLF6* gene could be crucial genomic markers for growth phenotypes of Chinese Datong yak breeds and this finding constitutes the first evidence of the biological role of *KLF6* CNVs in Chinese Datong yak breeds.

## 1. Introduction

The utilization of genomic selection in yak breeding has sped up the rate of genetic gain. Genomic divergence can be observed in several forms, including single nucleotide polymorphisms (SNPs), variable numbers of tandem repeats, transposable elements and structural variations such as deletions, duplications and inversions. Until recently, SNPs were considered the major form of genomic variation, accounting for much of the normal phenotypic variation that is observed. Notably, copy number variations (CNVs) encompass higher nucleotide content within the genome than SNPs and thus represent novel markers for the assessment of genetic diversity and evolution [1]. The use of advanced genomic technologies for CNV detection, which range from microarrays to the more recently development next-generation sequencing approach, is significantly contributing to the elucidation of genetic and phenotypic variation in domestic animals [2,3,4,5,6,7,8,9]. Diverse genomic characteristics can control quantitative traits through variations in gene expression levels [10], ultimately contributing to inter-individual phenotypic variability. Genomic copy number variations stretching from 1 kb to several Mb contain gains (duplication) or losses (deletions or null genotypes) compared with the designated reference genome sequence [11]. CNVs can affect gene expression and influence both phenotypic variability and evolutionary adaptation due to the wide range of copy numbers of a specific sequence within the genome [1,12,13] obtained by altering the gene dosage, transcript structure and regulatory polymorphisms [14,15].

Several recent studies have shown that CNVs comprise approximately 4.6% of the entire bovine genome [5,6,16,17]. The association of CNVs with phenotypic traits was recently utilized for the determination of feed conversion ratios in beef cattle [18], milk somatic cell scores in Holstein cattle [2,3,19] and quantitative traits in a Brown Swiss population [10]. A further study using qPCR in Chinese cattle revealed that CNVs in the genes MICAL-like 2 (*MICAL-L2)* [20], leptin receptor (*LEPR)* [21], myosin heavy chain 3 *(MYH3)* [22], potassium voltage-gated channel subfamily J member 12 (*KCNJ12)* [23] mitogen-activated protein kinase 10 (*MAPK10)* [24], phospholipase A2 group IID (*PLA2G2D)* [25] and cytochrome P450 family 4 subfamily A member 11 (*CYP4A11)* [26] are associated with gene expression and growth traits. On the other hand, Zhang et al. [13] 29] demonstrated associations for CNVs with genetic diversity between individuals [27] and susceptibility to a complex human disease [28].

The yak (*Bos grunniens*) is bred for survival in the extremely harsh environment of the Qinghai-Tibetan Plateau and adjacent alpine regions [29] and significantly impacts sociocultural factors and daily life by providing milk, meat, transportation (as the “boat of the plateau”), hair, draught power and fuel for Tibetans and nomads [30,31]. Yak breeds survive in the natural highland environment, which harbors native species and natural resources and graze on pasture to produce organic food (organic animal husbandry) [32,33]. Using next-generation sequencing (NGS) with read-depth methods, a recent study suggested that CNVs in Kruppel-like factor 6 (*KLF6*) are related to yak meat production and quality, which are economically significant traits that have been extensively considered for artificial selection in yak breeding programs [9]. *KLF6* belongs to the Kruppel-like family and is a ubiquitously expressed zinc finger transcription factor [34] that is primarily involved in cell proliferation, differentiation, development [35,36] and growth-related signal transduction [37]. Andreoli et al. [38] found that *KLF6* affects the postnatal mass and development of skeletal muscle [39,40] by regulating genes involved in the transforming growth factor beta 1 (*TGFB1*) signaling pathway [41]. Thus, Zhang et al. [9] postulated that *KLF6* CNVs might influence muscle development and thus the phenotypic traits of yak breeds through gene regulation and dosages. Further work is required to attain a more complete understanding and validate the importance of the detected CNVs of the *KLF6* gene found in yak breeds. However, the importance of the *KLF6* gene in relation to growth traits of Chinese domestic yak breeds has not been previously investigated. As a result, this research aimed to assess the prevalence of *KLF6* CNVs in five Chinese domestic yak breeds and to determine their association with growth and gene expression levels. Therefore, this study was hypothesized that the copy numbers types of the *KLF6* gene was negatively or positively associated with DNA copy number and the transcriptional level of the *KLF6* gene. The results suggest that CNVs of the *KLF6* gene could be crucial genomic markers for growth phenotypes of Chinese Datong yak breeds and this finding constitutes the first evidence of the biological role of *KLF6* CNVs in Chinese Datong yak breeds and indicates the novel possibility of using CNVs in the design of molecular markers for selective animal breeding programs, which will be necessary for the analysis of a larger variety of yak breeds in the future.

## 2. Materials and Methods

The study was conducted between September 2016 and December 2017 at the Key Laboratory of Yak Breeding Engineering of Gansu Province, Lanzhou Institute of Husbandry and Pharmaceutical Sciences and the data were obtained from yak breeding cooperatives of Gansu and Qinghai Province, China. All blood sample collection and body measurements were performed in strict accordance with the guide for the Care and Use of Laboratory Animals, Lanzhou Institute of Husbandry Animal and Pharmaceutical Sciences, China. Additionally, all the animals were slaughtered under anesthesia and all necessary efforts were made to minimize the risk of suffering. Thus, we were granted permission to perform this research study on yak and the legal certificate number was SCXK (Gan) 2014-0002.

### 2.1. Study Areas and Quantitative Traits

Chinese domestic yak breeds are currently maintained for the production of meat and milk and for genetic conservation. Therefore, the following five representative Chinese domestic yak breeds were selected: Polled (Datong Yak Farm, 101°70′E, 32°N, altitude 3200 m, Qinghai Province, China), Tianzhu white (102°02′–103°29′E; 36°29′–37°41′N, altitude 3000 m, Gansu Province, Tianzhu Tibetan Autonomous County, China), Plateau (35°34′11″N, 100°46′45″E, altitude 3700–4700 m, northern and southern Qinghai Province, China), Datong (Datong Yak Farm, 101°70′E, 32°N, altitude 3200 m, Qinghai Province, China) and Gannan (100°46′–104°45′E, 33°06′–35°43′N, altitude 3300–4400 m, Gansu Province, bordering Sichuan and Qinghai, China). The animals were healthy and assumed to not have any genetic relationships. All yaks were allowed to graze on natural pasture without any feeding supplementation under similar feeding and management conditions. The phenotypic data of the 387 individual Datong yaks (i.e., body weight, body height, body length, chest girth and cannon width) were collected at three stages of growth (i.e., 6 months (*n* = 222), 3 years (*n* = 72) and 5 years (*n* = 93) old) for association studies. The body measurements were collected following the guidelines described by Gilbert et al. [42].

### 2.2. Sample Collection, Deoxyribonucleic Acid (DNA) and Ribonucleic Acid (RNA) Isolation and cDNA Synthesis

Fresh blood samples were collected from the jugular vein of 507 yaks (387 Datong yaks, 30 Polled yaks, 30 Tianzhu yaks, 30 Gannan yaks and 30 Plateau yaks) into vacuum tubes (5 mL). For total RNA extraction, tissue samples containing lung, brain, spleen, skeletal muscle, heart, liver, kidney and adipose fat (except fetal adipose fat) were obtained from three 90-day-old fetuses (embryo) and three 3-year-old (adult) Datong yaks. In addition, 41 (*n* = 41) skeletal muscle samples from different adult Datong yaks were collected for both total RNA and genomic DNA studies. After collection, these blood and tissues samples were stored at −80°C for DNA and RNA isolation.

Genomic DNA (gDNA) from blood samples was purified using a Clot Blood DNA kit (CWBIO, Beijing, China) and genomic DNA longer than 10 kb was obtained using a E.Z.N.A. MicroElute DNA Clean-Up kit (Omega Bio-tek, Norwalk, GA, USA) following the manufacturer’s instructions. gDNA from skeletal muscles was extracted with a DNeasy^®^ Blood and Tissue kit (Qiagen^®^) following the manufacturer’s recommended standard procedure. Total RNA from the tissues was also isolated using a TRIzol reagent (TriPure Isolation Reagent, Roche, Carlsbad, CA, USA) and an RNeasy^®^ Blood and Tissue kit (Qiagen^®^), according to the manufacturers’ instructions. The concentration and quality of RNA and DNA were examined using a NanoDrop™ BioPhotometer 2000 (Thermo Fisher Scientific, Inc., Waltham, MA, USA) and through electrophoresis on ethidium bromide-stained 1% agarose gels.

For the accurate analysis of gene expression, complementary DNA (cDNA) was synthesized without contamination from genomic DNA by designing primers corresponding to exon regions spanning introns. In addition, cDNA was produced by reverse transcription from 1 μg of RNA using a PrimeScript^TM^ RT reagent kit with gDNA Eraser (Perfect Real Time) (TaKaRa Bio Inc., Shiga, Japan).

### 2.3. Primer Design and PCR Amplification

Primers were designed for the analysis of CNVs and gene expression using the National center for Biotechnology information (NCBI) primer-BLAST web tool (https://www.ncbi.nlm.nih.gov/tools/primer-blast/index.cgi?LINK_LOC=blastHome) [43] (Table 1). Polymerase chain reactions (PCRs) in a total volume of 25 μL, containing 50 ng gDNA or cDNA, 2X GoTaq^®^ Green Master Mix, 10 μM primers and nuclease-free water (ddH_2_O), were used to validate the amplification primers (Promega, Madison, WI, USA). The thermal cycling conditions were as follows: 95 °C for 2 min as the initial denaturation step followed by 35 cycles of denaturation at 95 °C for 1 min, annealing at 55–60 °C for 1 min and extension at 72 °C for 1 min, a final extension at 72 °C for 5 min and holding at 4 °C. The PCR products were loaded directly into 1% agarose gels or non-denaturing TBE polyacrylamide gels. The amplification primers were also checked by analysis of the melting curves obtained during the qPCR assay.

### 2.4. Analysis of Copy Number Variation and Expression of the KLF6 Gene

The primers designed for the analyses of copy number variation and gene expression are listed in Table 1. The relative CNVs and expression levels of the bovine *KLF6* gene were investigated in this study by qPCR. Bovine basic transcription factor 3 (*BTF3*) was used as a normal (two-copy or diploid) internal reference gene for genomic qPCR because neither CNVs nor segmental duplication have been described for this gene in the Database of Genomic Variants (https://www.ncbi.nlm.nih.gov/dbvar) [44]. Bovine glyceraldehyde-3-phosphate dehydrogenase (*GAPDH*) was selected as the reference gene in the gene expression analysis due to its ubiquitous expression in most tissues and cells [45]. Gene expression levels were evaluated by genomic qPCR using the Bio-Rad CFX 96™ Real-Time Detection System (Bio-Rad, Hercules, CA, USA). The qPCR was performed using SYBR^®^
*Premix Ex* Taq^TM^ II (Tli RNase H Plus, Nojihigashi 7-4-38, Kusatsu, Shiga 525-0058, Japan) in a 25-μL total reaction mixture containing 50 ng of gDNA or cDNA, 12.5 μL of SYBR *Premix Ex* Taq II (2X) and 10 pmoL of primers (TaKaRa Bio, Inc., Shiga, Japan). The thermal cycling conditions were as follows: one cycle of 95 °C for 1 min followed by 39 cycles of denaturation at 95 °C for 10 s, annealing at 60 °C for 30 s and extension at 68 °C for 10 s. During the qPCR thermal cycling, the primers were also examined by melting curve (dissociation) steps and using no-template control (negative control) reactions. The experiments were repeated three times and the mean values and standard deviations from the data were used for the statistical analyses.

### 2.5. Statistical Analysis

The threshold cycle (ΔΔC_t_) values were used to determine relative copy numbers [46]. Specifically, a DNA copy number was calculated using the average threshold cycle (ΔC_t_) from the three replicates and normalized against the reference gene *BTF3* using the formula 2 × 2^−ΔΔCt^ [47], where a copy number of two (diploid) is the normal DNA copy number [46]. The Ct values were converted to the nearest integer as described previously [20,21,22,23,24]. The quantitative mRNA expression levels of the target gene were determined using the threshold cycle 2^−ΔΔCt^ method [48]. The association between CNVs in the *KLF6* gene and growth traits in the Datong yak breed was assessed by one-way analysis of variance (ANOVA) using SPSS software (IBM SPSS 20, Hausmalt, CH, Switzerland). In the biostatistical model, various types of CNVs were classified as loss (copy number of 0 or 1), gain (copy number > 2) and normal (copy number of 2) relative to the reference gene *BTF3*, as described elsewhere [20,21,22,23,24]. The growth traits were fitted using the following analytical models: Y*_ij_* = μ + G*_i_* + D*_j_* + ε*_ij_*, where Y*_ij_* = number of records of copy number types *i* at age *j*, μ = the overall mean, G*_i_* = fixed effects of CNV type *i* (*i* = 1, 2, 3), D*_j_* = fixed effect of age *j* (*j* = 1, 2, 3) and ε*_ij_* = residual error. Moreover, the Bonferroni correction, which presents a modified significance criterion (*p*/*m*, where *m* is the overall number of independent statistical tests conducted on the given data and *p* is the significance level of 0.05), was utilized to avoid type I errors derived from multiple comparisons [20].

The diversity between yak breeds was determined in relation to the control sample using the log_2_ ratio (log_2_^2−ΔΔCt^) values and the types of CNVs were defined as gain (>0.5), loss (<−0.5) or normal (<|±0.5|) [20,21,22,23,24]. Pairwise comparisons and scatterplot analyses of the yak breeds were performed using GraphPad Prism software (version 5.00) and the values were obtained from the log_2_ ratio (log_2_^2−ΔΔCt^). The pairwise comparisons between breeds were tested using Tukey’s multiple comparison tests.

Pearson product-moment correlation was performed using R 2.15.0 software (WU Wien, Austria) to determine the relationship between the log_2_ ratio of the DNA copy number and the messenger RNA (mRNA) *KLF6* expression levels [49]*.* According to Stranger et al. [50], the correlation generated using log_2_ ratio signals yielded strong *r^2^* and *p* values; therefore, a correlation study was performed directly using data obtained from log_2_ ratios as described previously [24].

## 3. Results

### 3.1. Association of Gene CNVs with Growth Traits in Yak

We performed one-way analysis of variance (ANOVA) to determine the association between *KLF6* CNVs and growth traits of the Datong yak breeds. In the statistical analysis, the *KLF6* copy numbers were normalized against the reference gene *BTF3* in the 387 individual of Datong yaks. As described in Table 2, the loss CNV type of the *KLF6* gene was more highly correlated with growth traits, namely, body length (*p* = 0.0001), chest girth (*p* = 0.002) and body weight (*p* = 0.003), than the gain or normal types (*p* < 0.05) in five-year-old yak. In addition, the loss CNV type of the *KLF6* gene was significantly (*p* < 0.05) associated with increased body weight, chest girth and body length in 6-month-old Datong yak compared with the gain type (Table 2). However, the effects of the CNV type of the *KLF6* gene on growth traits were not significant in 3-year-old yak (Table 2).

### 3.2. Population Genetic Analyses of CNVs

As presented in Figure 1A, the CNVs of the *KLF6* gene in Tianzhu, Gannan and Plateau yak individuals exhibited increased copy number loss and increased Datong yak individuals had normal copy numbers, whereas more Polled yak individuals possessed copy number gains. Across all the breeds, the overall frequencies of the *KLF6* copy number (Figure 1B) classified as loss (copy number of 0 or 1), normal (copy number of 2) and gain (more than 2 copies) were 40%, 34% and 26%, respectively (Appendix A). Among these, 56.7% of Tianzhu yak, 56.7% of Plateau yak and 60% of Gannan yak showed only the loss CNV type, whereas 66.7% of Datong yak showed only normal copy numbers and 96.7% of Polled yak exhibited gains in copy numbers of the *KLF6* gene (Appendix A). Highly significant differences were observed in pairwise comparisons of Polled compared with Tianzhu, Gannan, Datong and Plateau yak (*p* < 0.0001) and with non-significant differences were found between the Datong and Gannan yak breeds (*p* < 0.05); however, no significant differences were observed in the comparisons of Datong versus Plateau, Datong versus Tianzhu, Tianzhu versus Gannan, Tianzhu versus Plateau, or Gannan versus Plateau yak breeds (Appendix A).

### 3.3. Analysis of the Expression Pattern of the KLF6 Gene in Different Tissues

The *KLF6* gene plays a prominent role in the regulation of cell differentiation and proliferation and muscle development. Nevertheless, the expression and biological function of the *KLF6* gene in yak have not been explored. Here, we elucidated the *KLF6* gene expression patterns in seven different tissues from fetal Datong yak and in eight different tissues from adult Datong yak (Figure 2A–C). The results indicated that *KLF6* showed differential expression between the two developmental ages and significantly (*p* < 0.05) higher expression of *KLF6* was observed in the lung, muscle, kidney and spleen in fetal yak (Figure 2A). In addition, a moderate mRNA expression level was observed in the heart, brain and liver tissues. In adult yak, the highest expression level of *KLF6* was detected in adipose, lung, muscle and spleen tissues, whereas moderate expression was observed in the brain and kidney and weakly expression was found in the liver and heart (Figure 2B). The comparison of the two developmental ages revealed significantly higher (*p* < 0.05) expression in the lung, muscle, kidney and spleen tissues of fetal yak (Figure 2C).

### 3.4. Correlation Analysis of CNVs and KLF6 mRNA Expression

Among 41 samples, the tested skeletal muscle was found to contain 14, 24 and 3 cases of the normal, gain and loss CNV types of the *KLF6* gene. As shown in Figure 3A, among the individual tested yak muscles, the CNV types varied from 1 to 7 copies, whereas the expression of the gene varied from 0.2- to 1.6-fold (Figure 3B). A Pearson correlation analysis showed that the DNA copy numbers and mRNA levels of *KLF6* were non-significantly (*p* = 0.08) and negatively (*r* = −0.28) correlated in adult skeletal muscle (Figure 3C). The comparison of CNV types and gene expression revealed that a normal copy number was more significantly (*p* < 0.05) associated with a higher mRNA expression level than the gain or loss types, whereas no difference was observed between the loss and gain CNV types (Figure 3D). These results underscored that the loss copy numbers of the *KLF6* gene were predominant for growth traits and were correspondent with the negative association with between DNA copy number and the transcriptional level of the *KLF6* gene (Figure 3C,D). These findings illustrated that the presence of at least two copies of the *KLF6* gene was associated with higher expression of this gene.

## 4. Discussion

In yak, CNV surveying has been restricted to the discovery of CNV regions (CNVRs) using aCGH [25], BovineHD Genotyping Bead Chip Array [51] and next generation sequence (NGS) [9]. However, the detection methods and platforms used for surveying CNVs are varied and linked to differences in the resolution power, frequency, sampling size, genomic coverage, CNV calling algorithms and other technological advantages and disadvantages [52,53]. However, previous studies indicated that NGS is a more precise platform than array-based methods [9]. The identification of false positives and negative CNVs from different sequencing results has been accomplished by qPCR and/or fluorescence in situ hybridization (FISH) [4,9,52]. Moreover, in various species of livestock, different CNVs were broadly validated by qPCR [4,9,51]. As such, the detection of CNVs and gene expression levels by qPCR is a very efficient method [8,26]. Gains and losses in gene copies within the genome are related to the gene dosage and the mutation of regulatory factors influences gene expression [54]. Our results demonstrate that the loss CNV type of the *KLF6* gene is significantly (*p* < 0.05) associated with the body weight, chest girth and body length of 6-month-old and 5-year-old yak; however, the majority of the phenotypes examined exhibited considerable variation in CNV types. Indeed, Dionyssiou et al. [55] observed that *KLF6* depletion resulted in enhanced myogenic differentiation, which indicates that the presence of 0 or 1 copies of *KLF6* impacts growth traits and thus implies that the position of *KLF6* in the CNVRs that encompass QTLs is associated with economic quantitative traits [4]. In contrast, works revealed by Xu et al. [20,21], Yang et al. [26] and Liu et al. [24] found that gains in the copy number of the *MYH3, MICAL-L1, CYP4A11* and *MAPK10* genes were associated with high body weight, chest girth, body length and height in Chinese cattle. In agreement with these findings, losses in copy numbers of the insulin (*INS*) gene were associated with increased body mass index and waist circumference in humans [56]. Disagreement with these findings, gains in copy numbers of *LEPR* and neuronal growth regulator 1 (*NEGR1*) gene were associated with a decreased body mass index, waist circumference and risk of abdominal obesity in human [56]. Various CNVs unquestionably exert significant effects on meat production and reproduction traits in cattle [57].

As outlined above, *KLF6* CNVs showed diverse distributions among the yak populations. Normal CNVs were more common in Datong yak, whereas CNV gains were more common in Polled yak and CNV losses were more common in Tianzhu, Gannan and Plateau yak populations. Genetic diversity in yak breeds may indicate long-term effects of artificial insemination, hybridization and introgression of yak breeds. These results further explain the frequencies of different copy-number types in five breeds with variations that could originate from the diversity of breeds kept in different environments [26,58]. Datong yak are the first artificially cultivated yak breed created by the mating of Huanhu yak cows with wild yak (male) for meat production and Polled yak were obtained from the breeding of Polled yak cows with horned yak bulls [31,59]. The Tianzhu, Gannan and Plateau yak populations are among the indigenous yak of China. Comparable to these results, the gain copy type of the *GBP2* gene is in higher frequency in yak compared to Chinese cattle breeds [60]. These results might be explained by Upadhyay et al. [61], who found that a discrepancy in CNV between different cattle populations could occur due to population history, such as a change in past effective population size, gene flow and selection processes.

Biochemically, *KLF6* gene plays a significant role in the regulation of cell differentiation and proliferation and muscle development [37]. These results showed that the *KLF6* gene is expressed at significantly higher levels in the lung, muscle, kidney and spleen compared with other yak tissues, which indicates that the *KLF6* gene might play a vital role in the development of the lung, muscle, kidney and spleen. Dionyssiou et al. [55] identified cellular processes that are cooperatively regulated by the *TGFB1* and *KLF6* genes. The high expression of the *KLF6* gene in fetal yak tissues might explain the initial detection of the *KLF6* gene in placental cells and its essential role in the regulation of the TATA box-less genes encoding pregnancy-specific glycoproteins (PSGs) in humans [62], which are involved in placental development and pregnancy maintenance [63]. These results also support the work conducted by Atkins and Jain [64], who reported that the *KLF6* gene is highly expressed in the placenta [61], neuronal tissue, hindgut, heart, lung, kidney and limb buds during mid gestation [36,65]. Our results strongly indicate that *KLF6* showed differential expression in different yak tissues and at different developmental stages [24].

Remarkably, *KLF6* expression in the skeletal muscle of Datong yak was negatively influenced by CNVs. Indeed, the negative trend between the copy number and mRNA level of *KLF6* hinted at a potential effect of this CNV on muscle development and this finding was confirmed by the finding that the knockdown of wild-type KLF6 (*wtKLF6*) increased tumor growth in nude mice while reducing overall growth and the expression of angiogenesis-related proteins [66]. These results suggest that the mRNA expression and DNA copy numbers of *KLF6* are negatively correlated due to gene dosage effects. Consistent with this study, Merla et al. [67] showed that changes are not always directly correlated to CNVs and are influenced by various factors, including the deletion size, altered chromatin structure, a dosage-compensation mechanism, or a combination of these factors. Similarly, Liu et al. [26] revealed that CNVs affect gene expression through gene dosage and gene-gene or gene-environment interactions. According to a study by Stranger et al. [50], changes in copy number could cause variations in gene expression. There is evidence that such relative copy variations can be both negatively [21,25] and positively [20,24] correlated with relative mRNA expression. Moreover, this result confirmed that the differential effects of CNVs on phenotype traits are dependent upon the tissue specificity of the gene [24]. The finding that *cis*-expression QTLs (eQTLs) detected in rat tissues directly overlap with CNVRs indicates that CNVs are major tissue-independent regulators of gene expression [68]. Stranger et al. [50] reported that approximately half of the CNV-related effects on gene expression are caused by gene disruption or disturbances in regulatory and other functional regions rather than by altering the gene dosage. Somerville et al. [69] suggested that the copy number has negative effects on gene expression due to disruption of coding sequences; however, phenotypes associated with deletion and duplication within gene regions are sensitive to the dose. Thus, these findings confirm that quantitative growth traits are influenced by *KLF6* CNVs by altering transcriptional levels of the gene. Notably, these results underlie the fundamental role of *KLF6* CNVs and lay the foundation for future research aiming to design a breeding program for yak populations. In addition, we suggest that the breeding design affects the production of meat and milk by yak breeds; therefore, *KLF6* CNVs could be incorporated into breeding objectives by breeders to fulfill the requirement of livestock keepers and consumers. This investigation represents a milestone in the development of a commercial yak production business because the loss CNV type of the *KLF6* gene could be used in future molecular marker-assisted selection in yak breeding. These observations illustrate that losses in the copy number of the *KLF6* gene prominently affect yak growth traits, in agreement with the negative correlation found between the mRNA expression levels and CNVs of the *KLF6* gene.

## 5. Conclusions

In summary, this study elucidated the *KLF6* CNV distribution among five Chinese domestic yak breeds. The expression levels of *KLF6* were mostly high during fetal development (embryo) and low in adult tissues but these followed the same patterns at both developmental ages, supporting a biological role for this gene in cell proliferation and differentiation, development, apoptosis and angiogenesis, particularly during fetal development. These results confirmed the significant function of *KLF6* CNVRs in determining the mRNA expression levels of this gene in *Bos grunniens* skeletal muscles and the expression of this gene affects quantitative growth traits in yak populations through the negative relationship between DNA copy numbers and gene expression. The loss CNV type was positively associated with economically important quantitative traits in Datong yak. This work provides the first observation of the biological role of *KLF6* CNVs in Chinese Datong yak breeds and might therefore provide a novel opportunity to utilize data on CNVs in designing molecular markers for the selection of animal breeding programs for larger populations of various yak breeds.

## Figures and Tables

**Figure 1 animals-08-00145-f001:**
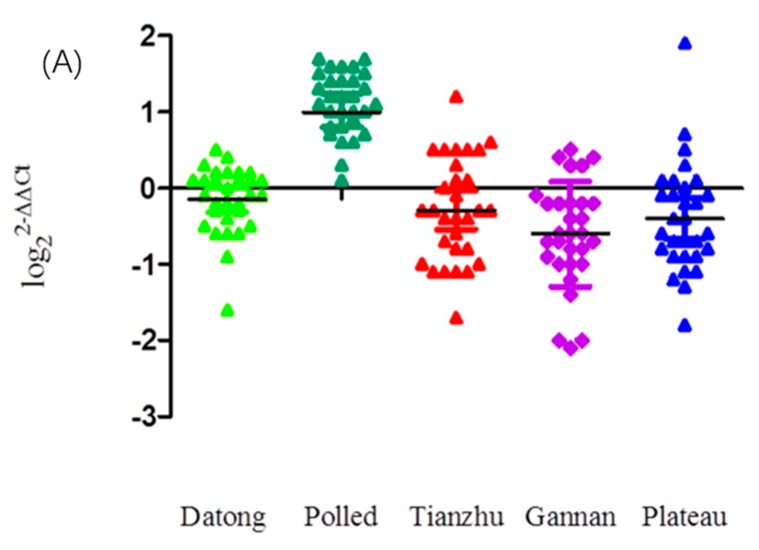
Distributions of CNVs of the *KLF6* gene in five yak breeds. (**A**) Scatterplot of CNV distributions of the *KLF6* gene in five yak breeds (*n* = 30). (**B**) Relative frequency of copy numbers classified as the loss (0 and 1), normal (2) and gain (>2) types in five yak populations.

**Figure 2 animals-08-00145-f002:**
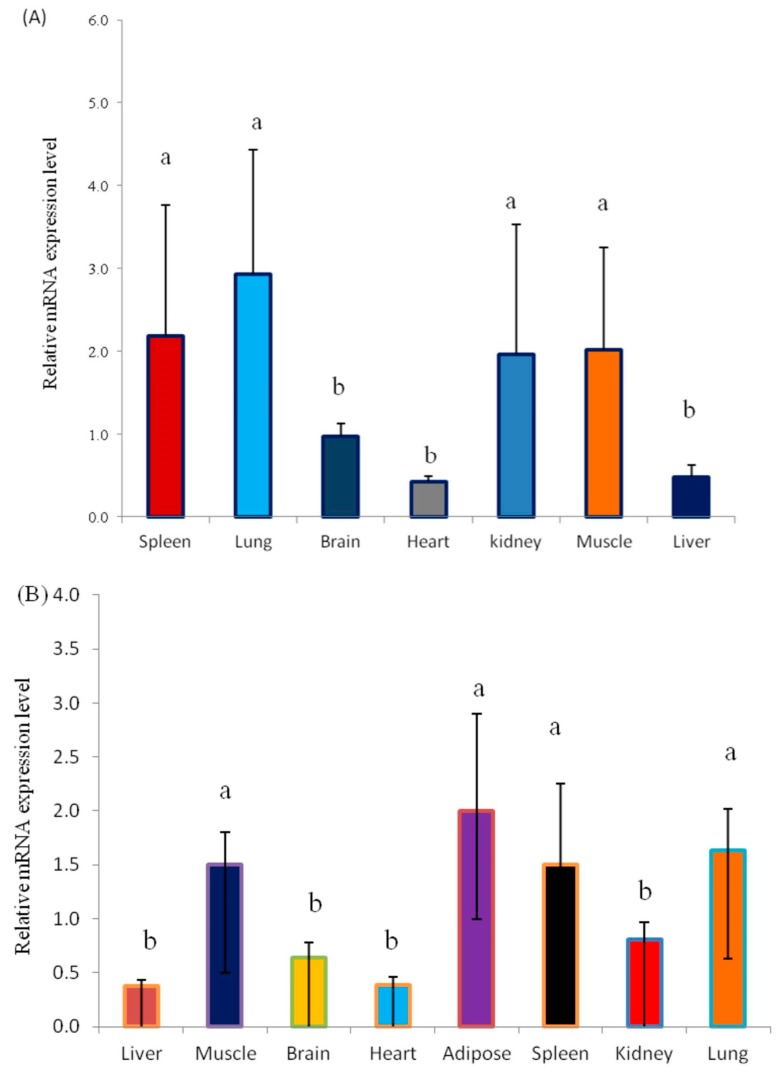
Expression profiles of the *KLF6* gene in Datong yak at two ages. (**A**) Expression pattern of the *KLF6* gene in fetal yak. (**B**) Expression pattern of the *KLF6* gene in adult yak. (**C**) Comparison of *KLF6* gene expression between the two ages. The error bars indicate the standard deviations (SDs). Different letters indicate significant differences (*p* < 0.05).

**Figure 3 animals-08-00145-f003:**
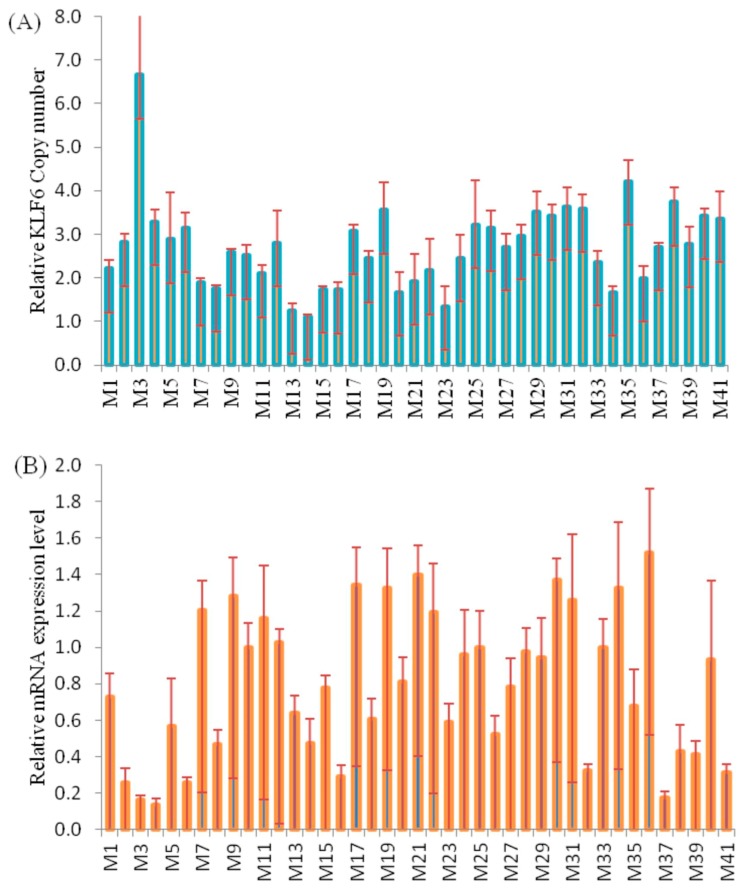
Analysis of the correlation between the log_2_ ratio of gene expression and CNVs of the *KLF6* gene in the skeletal muscle of adult Datong yak. (**A**) Relative CNV of the *KLF6* gene (*n* = 41, M1–M41). (**B**) mRNA *KLF6* gene expression level (*n* = 41, M1–M41). (**C**) Correlation between DNA copy number and mRNA *KLF6* gene expression level. (**D**) The impacts of DNA CNVs on *KLF6* mRNA gene expression level (≥1, *n* = 3; 2, *n* = 14; 3, *n* = 17; ≥4, *n* = 7) were evaluated between different CNV types. The error bars show the standard deviations (SDs). Different letters indicate a significant difference (*p* < 0.05).

**Table 1 animals-08-00145-t001:** qPCR primers designed based on bovine genomic sequences.

Level	Accession No.	Gene Name	Primer Sequence (5′–>3′)	Product Length (bp)	Annealing Temperature (°C)
DNA	XM_005901807.2	*KLF6*	F: ATGCTCATGGGAAGGGTGTG	82	57.45
			R: CTTGGCACCAGTGTGCTTTC		57.45
	NM_001034034.2	*BTF3*	F: AACCAGGAGAAACTCGCCAA	166	55.40
			R: TTCGGTGAAATGCCCTCTCG		57.45
mRNA	NM_001035271.3	*KLF6*	F: GTGACAGGTGTTTCTCCAGG	91	57.45
			R: TTTTAGCCCGCAGGAGTTGT		55.40
	NM_001034034.2	*GAPDH*	F: AATGAAAGGGCCATCACCATC	204	55.85
			R: GTGGTTCACGCCCATCACA		60.00

**Table 2 animals-08-00145-t002:** Analysis of the association between copy number variations (CNVs) in the *KLF6* gene and growth traits in Datong yak breed.

Age	Growth Trait	CNV Type (Mean ± STD)			*p*-Value
		Loss (*n* = 70)	Normal (*n* = 71)	Gain (*n* = 81)	
6 months (*n* = 222)	Body height (cm)	93.9 ± 2.4 ^a^	90.9 ± 3.6 ^b,c^	91.2 ± 4.3 ^b^	0.0001
	Body length (cm)	92.3 ± 1.8 ^a^	89.2 ± 3.8 ^b^	89.5 ± 4.3 ^b^	0.0001
	Chest girth (cm)	122.2 ± 4.8 ^a^	119.7 ± 6.8 ^b^	118.7 ± 6.0 ^b^	0.002
	Cannon width (cm)	12.0 ± 0.24	12.1 ± 0.4	12.1 ± 0.32	0.100
	Body weight (kg)	93.9 ± 9.2 ^a^	89.2 ± 13 ^b^	87.6 ± 10.7 ^b^	0.002
3 years (*n* = 72)		Loss (*n* = 25)	Normal (*n* = 31)	Gain (*n* = 16)	
	Body height (cm)	104.0 ± 3.2	104.4 ± 3.2	104.4 ± 3.5	0.89
	Body length (cm)	114.0 ± 4.9	110.4 ± 19.3	114.1 ± 4.9	0.53
	Chest girth (cm)	147.2 ± 6.9	148.8 ± 6.4	147.2 ± 6.3	0.59
	Cannon width (cm)	16.4 ± 0.51	16.5 ± 0.51	16.5 ± 0.52	0.92
	Body weight (kg)	176.0 ± 20.6	174.6 ± 36.4	175.9 ± 17.7	0.98
5 years (*n* = 93)		Loss (*n* = 4)	Normal (*n* = 4)	Gain (*n* = 85)	
	Body height (cm)	111.8 ± 2.1	112.0 ± 0.82	109.3 ± 3.1	0.065
	Body length (cm)	133.0 ± 10 ^a^	124.9 ± 1.2 ^b^	123.4 ± 3.5 ^b^	0.0001
	Chest girth (cm)	166.8 ± 15.5 ^a^	157.0 ± 10 ^a,b^	155.3 ± 5.3 ^b^	0.002
	Cannon width (cm)	16.4 ± 2.2	16.5 ± 0.5	16.5 ± 0.9	0.43
	Body weight (kg)	257.4 ± 83.1 ^a^	220.2 ± 29.7 ^b^	211.3 ± 20.7 ^b^	0.003

Note: Different letters (^a, b, c^) indicate significant differences (*p* < 0.05); Standard deviation (STD).

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
