# Peer review of "Copy Number Variations of KLF6 Modulate Gene Transcription and Growth Traits in Chinese Datong Yak (Bos Grunniens)"

_animals, 2018, doi:10.3390/ani8090145_

Round 1
Reviewer 1 Report
The authors present a paper on copy number variations of KLF6 gene and the influence of its gene expression on the growth traits in Chinese domestic yak breeds.
The paper could be interesting for the readers but there are some points of concern that make the paper not acceptable in the present form.
Simple summary: line 15 it could be better to replace “with growth and transcription” with “with growth and gene expression”;
-line 21: it could be better to write “and gene expression level”;
-keywords: it could be better to avoid the use of words already included in the title;
-introduction, line 45: please delete the first sentence.
-line 80 please replace “methods, a recent suggested” with “methods, a recent study suggested”;
-line 93, please replace “and gene expression” with “and gene expression levels”. Beside replace “These results” with “The results”;
Material and methods
General commet: it is not easy to understand the experimental design. I have some doubt about it:
-the authors collected samples from 507 yaks from 5 different breeds. What are the reasons to have an unbalanced sample within the 5 breeds? Please clarify.
I understood the the authors collected:
507 blood samples from yaks of 5 breeds
6 samples for RNA analysis from 6 Datong yak
41 skeletal muscle from different yaks. What breeds?
I think that the authors must very well explain the sample structure. Otherwise it is very hard to understand all the obtained results. I have the feel that only a part of 507 samples were used…. Please clarify this aspect.
Please clarify why the authors collected samples in embryo at 90 days old. Is there some biological reason?
Line 130 please replace “for Deoxyribonucleic acid (DNA) and Ribonucleic acid (RNA) isolation” with “for DNA and RNA isolation”.
Moreover I have the feel that all the study is about Datong yak. The authors supplied only very few information about the other 4 breeds. For this reason also the title of the paper should be changed. Just a suggestion: Copy Number Variations of KLF6 Modulate gene expression and Growth Traits in Chinese Datong Yak (Bos grunniens) Breed.
Line 160, replace “Table 2” with “Table 1”.
Results
It could be better to move to material and methods section the text from line 206 to line 209.
At the end of paragraph 3.2 please use italic style for “vs”.
In the discussion section there are some part of the text that need to be summarized or moved to introduction section, i.e. lines 285-296, 302-313, 349-363.
Moreover the paragraph about the diversity (lines 314-327) is out the the aim of the paper: I suggest to the authors to delete it.
In the conclusion section (lines 383-386) it is speculative to refer about 5 breeds: as I already wrote the big part of the paper is only about Datong breed.
Author Response
We really appreciated your recognition to our work, and provided many reasonable suggestions to make our manuscript be closer to animals’ publication criteria. Actually, we have revised it carefully according to your recommendations and the changes and responses are as follows:

Reviewer 2 Report
The manuscript entitled “Copy Number Variations of KLF6 Modulate Gene Transcription and Growth Traits of Chinese Domestic Yak (Bos grunniens) Breeds” detected the KLF6 CNVs in five yak breeds but the authors investigated their association with growth traits and the expression pattern only in Datong breed. So, I suggest deleting “BREEDS” from the title. Some specific issues are indicated below.
Simple Summary
(L15) Remove “with growth”.
Abstract
(L30) Remove “sequencing”.
(L32) “…thereby contributes to development, differentiation, and proliferation”. Are you talking about cell differentiation and cell proliferation? What kind of development are you talking about (organism, organ, tissue, …)? Please, rewrite the sentence.
Keywords
Do not repeat the words used in the title.
Introduction
(L59-59) “CNVs are present in the form of many deletions, 58 duplications, insertions, inversions and translocations [13]”. Please, provide a modern CNV definition. Translocations, insertions, and inversions do not necessarily modify the copy number of DNA segments.
(L71-73) Delete or rewrite the following sentence: “In addition, Zhang et al. [9] demonstrated associations for CNVs with genetic diversity between individuals [31] and susceptibility to a complex human disease [32]”. There is no connection with the information previously presented because you are presenting studies with human (31 and 32 references).
Material and Methods
(L112-117) Please delete “yak” after the name of each breed: “Polled yak … Tianzhu white yak … Plateau yak … Datong yak … and Gannan yak”
(L120-121) Where the traits (body weight, body height, body length, chest girth and cannon 120 width) collected only of Datong yak? Please describe in “material and methods” the traits collected from Polled, Tianzhu white, Plateau, and Gannan yak breeds too.
(L128-129) “In addition, 41 (N=41) skeletal muscle samples from adult yak were 128 collected for both total RNA and genomic DNA studies”. Please, describe the yak breeds.
(L145) “Based on the bovine genomic sequences”. Please, provide the name of databank used in this study (NCBI?) and its reference.
(L145) “pair of primers was constructed”. You constructed four pairs of primers. Please, rewrite the sentence including all the primers described in Table 1.
(L149) “50 ng/μL gDNA and cDNA”. It is too much DNA. I think that you used only 50ng instead of 50 ng/μL and you used gDNA OR cDNA. Please, correct it.
(L150) “2x GoTaq® Green Master Mix”. Please provide the manufacturer.
(L157) Provide the annealing temperature of each primer in Table 1.
(L160) Replace “Table 2” by “Table 1”.
(L163) Provide a correct reference to Database of Genomic Variants instead [4].
(L168) 50 ng of gDNA OR cDNA
(L188-190) “The effects associated with sex, breed and location were not matched in the linear model because the preliminary statistical analysis indicated that these effects did not have a significant influence on the variability of the traits in the study subjects [24-28]”. What do you mean? Have you performed the preliminary statistical analysis, or it is a conclusion only based on the literature (24-28)?
(L207) Datong yak BREED.
Results
(L219) Table 2. The sample size of 5 years with deletion (N=4) and Normal (N=4) is too small to perform a comparison.
(L275) Fig3. How can you explain that animals with the same relative KLF6 copy number (Figure 3A, e.g. M40 and M41 samples) showed significant differences in relative mRNA expression level (Figure 3B, e.g. M40 and M41 samples)? The opposite is also true (e.g., M3 and M4). Maybe you should consider the sex, age and location to improve your discussion about it because in the “discussion” section you provide many references to try to explain it (L347-359), but you don’t provide a hypothesis.
Discussion
(L287) “methodological detection methods” is bad English.
(L303-306) “Liu et al. [48] reported that multiple genes found within CNVRs overlap with production and reproduction quantitative trait loci (QTLs). Additionally, CNVRs overlap with genes linked to parasite resistance, immunity response, body size, fertility, and milk production [8]”. It is not relevant to your discussion. It is true that CNVRs overlap with genes linked to all trait described in cattle, including production and reproduction QTL. You should focus the discussion only on growth traits.
(L306-313) Delete. It is not relevant.
(L318-320) “These results further explain that the variations in the frequencies of different CNV types in five breeds could originate from the diversity of DIFFERENT ENVIRONMENT in which the breeds are maintained”. The effect associated with location was not matched in the linear model. Please, explain it better.
(L329) “to development, differentiation, proliferation, and signal transduction [41]”. Rewrite this sentence.
Conclusions
(L381) What do you mean with “Surprisingly”?
Author Response

(The authors gave the same response as above.)

Round 2
Reviewer 1 Report
First of all I want to thank the authors for their efforts to improve the manuscript that in the present revised version is more readable and useful for the readers.
I have the opportunity to highlight just three typos that could be amended also in the editing step:
-line 138 it could be better to replace "The density and quality of RNA...." with "The concentration and quality of RNA...."
-Table 1, annealing temperature column, please replace "60" with "60.00" and "55.4" with "55.40"
-In the title of table 2, please replace "breeds" with "breed".